# Current State of Knowledge Regarding WHO High Priority Pathogens—Resistance Mechanisms and Proposed Solutions through Candidates Such as Essential Oils: A Systematic Review

**DOI:** 10.3390/ijms24119727

**Published:** 2023-06-04

**Authors:** Mirabela Romanescu, Camelia Oprean, Adelina Lombrea, Bianca Badescu, Ana Teodor, George D. Constantin, Minodora Andor, Roxana Folescu, Delia Muntean, Corina Danciu, Olivia Dalleur, Stefan Laurentiu Batrina, Octavian Cretu, Valentina Oana Buda

**Affiliations:** 1Doctoral School, “Victor Babeş” University of Medicine and Pharmacy, 2 Eftimie Murgu Street, 300041 Timisoara, Romania; mirabela.romanescu@umft.ro (M.R.); adelina.lombrea@umft.ro (A.L.); bianca.badescu@medicis.ro (B.B.); ana.teodor@umft.ro (A.T.); george.constantin@umft.ro (G.D.C.); 2Faculty of Medicine, “Victor Babeş” University of Medicine and Pharmacy, 2 Eftimie Murgu Street, 300041 Timisoara, Romania; andor.minodora@umft.ro (M.A.); folescu.roxana@umft.ro (R.F.); muntean.delia@umft.ro (D.M.);; 3Faculty of Pharmacy, “Victor Babeş” University of Medicine and Pharmacy, 2 Eftimie Murgu Street, 300041 Timisoara, Romania; 4OncoGen Centre, County Hospital ‘Pius Branzeu’, Blvd. Liviu Rebreanu 156, 300723 Timisoara, Romania; 5Multidisciplinary Research Center on Antimicrobial Resistance, “Victor Babes” University of Medicine and Pharmacy, 2 Eftimie Murgu Square, 300041 Timisoara, Romania; 6Research Center for Pharmaco-Toxicological Evaluation, “Victor Babes” University of Medicine and Pharmacy, 2 Eftimie Murgu Square, 300041 Timisoara, Romania; 7Louvain Drug Research Institute, Université Catholique de Louvain, Avenue Emmanuel Mounier 73, 1200 Brussels, Belgium; 8Faculty of Agriculture, University of Life Sciences “King Mihai I” from Timisoara, Calea Aradului 119, 300645 Timisoara, Romania; 9Ineu City Hospital, 2 Republicii Street, 315300 Ineu, Romania

**Keywords:** multidrug-resistant strains, nosocomial infections, vancomycin-resistant *Enterococcus faecium*, clarithromycin-resistant *Helicobacter pylori*, fluoroquinolone-resistant *Campylobacter* spp., cephalosporin-resistant, fluoroquinolone-resistant *Neisseria gonorrhoeae*, fluoroquinolone-resistant *Salmonellae*, methicillin-resistant, vancomycin-resistant *Staphylococcus aureus*

## Abstract

Combating antimicrobial resistance (AMR) is among the 10 global health issues identified by the World Health Organization (WHO) in 2021. While AMR is a naturally occurring process, the inappropriate use of antibiotics in different settings and legislative gaps has led to its rapid progression. As a result, AMR has grown into a serious global menace that impacts not only humans but also animals and, ultimately, the entire environment. Thus, effective prophylactic measures, as well as more potent and non-toxic antimicrobial agents, are pressingly needed. The antimicrobial activity of essential oils (EOs) is supported by consistent research in the field. Although EOs have been used for centuries, they are newcomers when it comes to managing infections in clinical settings; it is mainly because methodological settings are largely non-overlapping and there are insufficient data regarding EOs’ in vivo activity and toxicity. This review considers the concept of AMR and its main determinants, the modality by which the issue has been globally addressed and the potential of EOs as alternative or auxiliary therapy. The focus is shifted towards the pathogenesis, mechanism of resistance and activity of several EOs against the six high priority pathogens listed by WHO in 2017, for which new therapeutic solutions are pressingly required.

## 1. Introduction

### 1.1. Antibiotics and Antimicrobial Resistance

Antibiotics (ABs) are common agents used in healthcare to treat and prevent potentially fatal bacterial infections [1]. Their introduction into clinical practice was arguably the greatest discovery of modern medicine in the twentieth century [2]. Yet, ABs have been around for millennia, as many of them are compounds synthetized by microorganisms to protect themselves and dominate different habitats [3]. While humans started exploiting the power of ABs as early as 1550 BC, it was not until circa 100 years ago that we managed to understand, synthesize and purify ABs, salvarsan and penicillin being the promoters of the AB era [1]. Mishandling ABs has resulted in the rapid expansion of antimicrobial resistance (AMR). Even though AMR is a naturally occurring process, it was first identified 50 years ago, when *Staphylococcus aureus* began to develop penicillin resistance [4]. AMR is commonly associated with the presence of AB-resistant genes (ARGs) in the bacterial genome [5]. Given that bacteria are able to pass on ARGs through vertical or horizontal gene transfer [6], there is clear evidence that prolonged exposure to ABs can easily turn non-resistant bacteria into resistant ones [7]. 

### 1.2. Main Determinants of AMR

Given that AMR is linked to the emergence of multidrug resistant (MDR) and extensive drug resistant (XDR) pathogens [8], it represents a serious global threat of growing concern that affects humans, animals and the environment [9]. According to the World Health Organization (WHO), AMR is annually responsible for the deaths of at least 700,000 people worldwide and the death toll could reach 10 million by 2050 [10]. There are several reasons why AMR occurs: (1) inappropriate prescribing of ABs; (2) dispensing ABs without a prescription; (3) poor AB regulations and lack of surveillance of resistance development; (4) excessive use of ABs in food-producing animals; (5) limited decontamination of wastewater; and (6) lack of research on new ABs [8,11]. 

Initial, therapy errors such as misdiagnosing the infection aetiology, choosing an incorrect dosage, overly extending the duration of treatment, overlooking the recommended guidelines regarding first line ABs or prescribing ABs without a clear clinical indication all raise the risk of developing AMR [11,12]. These mistakes might be due to gaps in knowledge; fear of complications from infections; concerns of not meeting perceived patient expectation; financial benefits or incentives, and misleading advertising from the industry [13]. Poor regulations in AB use contribute to medication abuse and the occurrence of AMR. Dispensing ABs without a medical prescription is associated with inappropriate drug choice, wrong dosage, shorter treatment course and increased risk of adverse drug reactions [14]. Unregulated overuse of ABs in hospitals has increased the rates of resistance in nosocomial infections and thus cross-transmission [8]. Confronting MDR bacteria has led many hospitals to introduce antimicrobial stewardship programmes to monitor antimicrobial use and identify ways to reduce development and transmission of AMR [15]. However, just reducing the consumption of ABs and not working on socio-economic factors will not have the desired impact on the prevalence of AMR [16].

Antibiotic use in the animal industry has been found to be a key element in the development of AMR. Given that ABs might stimulate the intestinal synthesis of vitamins and lower the competition for nutrients between host and bacteria, sub-lethal doses of ABs are commonly used as growth promoters for food-producing animals [17]. In developed countries, 60–80% of ABs are also given to animals, especially poultry, pigs and cattle [18], with penicillins, tetracyclines and sulphonamides having the highest AMR rates [19]. In addition, the use of ABs as preservatives in meat [20] or heavy metals (such as copper and zinc) as growth promoters led to a massive increase in AMR [17]. Moreover, ABs are able to reach the environment via human and animal excretions, improper disposal of unused drugs and waste streams from the production line [21]. If micropollutants—including ABs and microorganisms—are not successfully removed, they represent an important source of soil and water contamination with ARGs [22]. 

Researchers are unable to keep the pace of finding new ABs in the face of emerging MDR strains. Starting with the discovery of penicillin, many ABs discoveries were serendipitous, made by empirical screening [23]. Nevertheless, despite technological progress, the last 25 years have not seen a breakthrough in the development of novel antibacterial drug classes [24] and there is a critical need for drugs targeting Gram-negative ESKAPE pathogens [25].

### 1.3. How AMR Is Addressed Globally and at European Level

In 2015, WHO developed an action plan to combat AMR which includes, among other objectives, the reduction of AB use in humans and animals [26], followed in 2017 by the publication of a list of bacteria that urgently need the development of new ABs, dividing them into three classes based on their healthcare burden. The following bacteria were classified as having critical priority (priority 1): *Acinetobacter baumannii*, *Pseudomonas aeruginosa* and *Enterobacteriaceae*. High priority pathogens (HPPs) were categorized as priority 2: *Enterococcus faecium*, *Staphylococcus aureus*, *Helicobacter pylori*, *Campylobacter* spp., *Salmonellae* spp. and *Neisseria gonorrhoeae* [27]. In May 2022, WHO issued a review of ABs in development worldwide. Since their first analysis in 2017, 12 new ABs have been approved; however, most were derivatives of existing classes, where resistance mechanisms have been established. At the moment, 27 ABs targeting WHO priority pathogens are in clinical development with 13 having confirmed activity against at least one of the critical Gram-negative bacteria [28]. The incoming United Nations General Assembly in 2024 aims to identify clear and practical steps to address AMR [29].

The global AMR situation is briefly presented by One Health Trust’s Resistance Map [30]. However, this database only covers the resistance of a few HPPs and some ABs (Table 1), whereas the reality might be more severe. On the other hand, the AMR picture in Europe is presented by the Surveillance Atlas of Infectious Diseases [31,32]. Here, although the situation varies broadly from country to country, the highest prevalence seems to be in the south-eastern region (Appendix A, Table A1). Given that both the European and the global databases lack reports from several countries, the real magnitude of AMR is difficult to grasp.

### 1.4. Essential Oils as Potential Tools against AMR

Essential oils (EOs), also called volatile oils, are complex mixtures of tens of lipophilic, volatile compounds, at different concentrations [33]. Generally, two or three major components are present in rather high concentrations (20–70%), while other compounds are present in trace amounts [34]. The main constituents of EOs are terpenes—compounds derived from isoprene units that often have several chemical functionalities, such as alcohol, phenol, aldehyde, ketone, ether and hydrocarbon groups [35,36]. EOs represent secondary metabolites produced by aromatic plants as a protective mechanism against predators, microorganisms or austere weather conditions [37]. Many parts of the plant are able to produce EOs, which can be then extracted using methods such as solvent extraction (solvent, subcritical water, supercritical CO_2_), distillation (hydrodistillation, steam distillation, hydrodiffusion), solvent-free microwave extraction and combined methods [38]. The applications of EOs range from aromatherapy and perfume production to food industry and animal nutrition [39].

In medical practice, EOs have been reported to have antimicrobial, antioxidant, anti-inflammatory, analgesic, antiemetic and cancer chemo-protective activities [40,41]. Some EOs also exhibit cytotoxic (against bacteria, viruses, fungi, protozoa, parasites and mites), allelopathic and insect repellent and insecticidal activities, thus they could be exploited as alternative strategies in a variety of industries [40]. Certain EOs are effective against pathogens of public health interest. In a previous review paper, we have presented that several EOs exhibit in vitro antimicrobial activity against the WHO priority 1 list of pathogens [42]. Notably, it is generally considered that EOs are more active against Gram-positive than Gram-negative bacteria [40]. Interestingly, unlike classic ABs, EOs exhibit good activity against pathogenic bacteria, while showing a lower effect on beneficial bacteria in the gut, such as *Lactobacillus* or *Bifidobacterium* [43]. Thanks to developments in pharmaceutical formulations, EOs can be loaded in carriers such as nanoparticles which significantly improve their bioavailability and stability [44]. However, the general population can also benefit from the effects of antimicrobial EOs through the use of spices [45].

The chemical functionalities responsible for antimicrobial activity are generally phenols and aldehydes, while a high proportion of esters, ketones and terpene hydrocarbons result in weak or no effect [46]. Thanks to their hydrophobicity, EOs can inhibit bacterial growth by degrading membrane proteins and increasing cell permeability [46,47]. They can interfere with the expression of genes encoding efflux pumps (tetA, tetK, pmrA, norA, blaTEM, blaOXA-23) in a variety of microorganisms [48]. Proton pumps can also be affected by EOs, resulting in reduced membrane potential and ATP depletion [49]. Moreover, EOs can hinder biofilm formation and disrupt quorum sensing. Thus, they alter cell-to-cell communication and interfere with gene expression regulation—crucial adaptive measurement in hostile environments [50].

The two most common in vitro techniques used to assess the antimicrobial activity of EOs are the agar diffusion method (paper disc or well) and the dilution method (agar or liquid broth) [51]. The agar diffusion methods are one of the most convenient techniques, in terms of price and methodology. In the agar well diffusion method, an agar plate is inoculated with a pathogenic microorganism through the spreading plate approach (an exact volume of the microbial solution is spread over the surface of agar, through a glass diffuser). A well or hole is aseptically made with a sterile cork borer of diameter 6–8mm, flooded with the tested solution (e.g., extract) and incubated at optimal temperature and conditions. The tested solution will diffuse progressively through the agar medium, thus inhibiting the microorganism growth. Later, the diameter of the inhibition zone will be measured. In the agar disc diffusion method, a filter paper disc containing the test solution is placed on the agar medium and then inoculated with the tested strains [51]. Overall, the agar diffusion techniques make it possible to easily test several extracts/substances against various microorganisms, although they are unable to highlight the minimum inhibitory concentration (MIC) or the ability of a substance/extract to inhibit or kill a microorganism [51].

In the dilution method a series of successive dilutions of concentrated solutions of microbial strains are prepared to accurately count the viable cells within a culture (bacterial, fungus or viruses). Each diluted sample is added to a liquefied agar medium, poured into a petri dish and solidifies, holding the microorganisms within its matrix. It is possible to count the microorganisms with precision as they disperse across the agar plate. This method is used to determine the MIC of a substance, as well as its ability to kill or inhibit the development of the tested strains. Moreover, it is the reference for antimicrobial susceptibility testing [51].

Bacterial resistance is generally defined using the minimum inhibitory concentration (MIC) and minimum bactericidal concentration (MBC) of the selected AB. However, the reported values for MIC and MBC are extremely divergent, possibly as a result of the wide diversity of bacterial strains, methodological dissimilarities and study design variations [52]. Notably, a number of variables related to the variability of the extract (the part of plant used, extraction method, etc.) also have an impact on EOs’ antimicrobial activity [53,54].

The aim of this review is to investigate the current level of knowledge regarding the six WHO HPPs: vancomycin-resistant *E. faecium* (VREF); clarithromycin-resistant *H. pylori* (CRHP); fluoroquinolone-resistant *Campylobacter* spp. (FRC); cephalosporin-resistant, fluoroquinolone-resistant *N. gonorrhoeae* (CRNG, FRNG); fluoroquinolone-resistant *Salmonellae* spp. (FRS); and methicillin-resistant, vancomycin-intermediate and -resistant *S. aureus* (MRSA, VISA, VRSA). To our knowledge, there have not been any recent publications that systematically present the antimicrobial activity of EOs against WHO priority 2 pathogens. Thus, our two main objectives are to present (1) HPPs’ key resistance mechanisms and (2) the latest EOs that exert antimicrobial activity on the aforementioned pathogens, with an emphasis on the AB-resistant strains.

## 2. Methods

### 2.1. Review’s Objective and Design

The aim of this review was to investigate the current level of knowledge regarding the six WHO HPPs (priority 2 list). This study was designed as a follow-up of a previous paper that addressed WHO critical priority pathogens (priority 1 list) [42]. Each of the six bacteria were discussed in terms of: pathogenesis, mechanism of AMR (with an emphasis on ABs indicated by WHO as increasingly inefficacious) and EOs as potential treatment candidates (with an emphasis on EOs that are effective against resistant strains). A working team was set up with members having experience in the area of interest: microbiology, clinical laboratory, clinical pharmacy, botany and pharmacognosy.

### 2.2. Literature Search Strategy

A flow diagram of screened records is provided in Figure 1. We conducted an independent search for each bacterium on three electronic databases (PubMED, Web of Science, Elsevier’s Scopus) and one web search engine (Google Scholar) from April 2022 to September 2022. The primary keyword combination used to perform the search was the following: WHO critical pathogen strain AND AB resistance AND essential oils (e.g., *Enterococcus faecium* AND vancomycin-resistant AND essential oils). Searches were conducted separately in each database and, after removing the duplicates, the records were exported to the citation software. Articles included in the study were those that: (1) focused on WHO priority 2 strains (primarily aiming for MDR strains); (2) investigated the activity of EOs against pathogens; (3) were published in the last 5 years; and (4) were written in English. The articles that failed to meet the inclusion criteria were excluded from the study, as well as the ones that fell under one of the following categories: book chapter, abstract, short communication, technical note, letter. Publications that appeared to be methodologically flawed and provided insufficient details or confusing outcomes were also dismissed.

### 2.3. Data Extraction

Pertinent articles were closely evaluated by reviewers and the following data were extracted: (1) study team and year of publication; (2) bacterial strain under investigation (including its resistance, if mentioned); (3) EOs/pure phytocompound proposed as having antimicrobial activity; method(s) of assessing the antimicrobial activity; and (4) main results. Any disagreements or queries were settled by a researcher with expertise in microbiology (D.M.).

## 3. Results and Discussion

### 3.1. Enterococcus Faecium, Vancomycin-Resistant

*Enterococcus* spp. are Gram-positive bacteria that are part of human faecal microbiota, as commensals of the intestinal tract [52]. However, some strains, particularly *Enterococcus faecalis* and *Enterococcus faecium*, may cause nosocomial outbreaks and opportunistic infections in hospitalized patients [53]. While most enterococci infections are caused by *E. faecalis* (nearly 90%), *E. faecium* has a higher rate of AB resistance, especially in case of bloodstream infections [54]. Plasmid transfer and homologous recombination, mediated by insertion sequence elements, are two important mechanisms through which *E. faecium* manages to escape effective therapy [55]. A series of genes (*ace*, *acm*, *scm* and *ecb*) code for a subset of adhesion factors that mediate the pathogen’s initial attachment. Aggregation substances, such as Enterococcal surface protein, favour *E. faecium*’s ability to form biofilms. Moreover, exoenzymes, including gelatinase, hyaluronidase and cytolysin, are secreted externally and can damage host cells by triggering inflammation [56].

Until recently, vancomycin was considered the gold standard in the therapy of beta-lactam resistant Gram-positive cocci. However, AMR is rising: in the United States, 30% of enterococci are vancomycin-resistant [57], while in some European countries the proportion of VREF is even higher (Table 1). VREF carried by patients is commonly shared within the hospital environment, as it may persist despite standard cleaning; thus, patients can acquire VREF if they are admitted to a room previously occupied by a VREF-positive patient [58]. VREF can be detected by identifying at least one of nine resistance genes—*vanA*, *B*, *C*, *D*, *E*, *G*, *L*, *M* and *N*—present in the mobile genetic elements (*vanA*, *vanB*) or located on the chromosome (*vanC*) [59]. The most predominant genotypes are *vanA* and *vanB*. *vanA E. faecium* is highly resistant to vancomycin and teicoplanin, while *vanB E. faecium* shows high resistance to vancomycin and susceptibility to teicoplanin [60]. Although linezolid has been considered a good alternative in treating VREF infections, resistance to this drug is gradually increasing, with G2576 T mutation within the 23S rDNA being one of the major resistance mechanisms [57]. Teicoplanin or last-line ABs such as daptomycin, tigecycline or quinupristin/dalfopristin remain an option for some VREF infections, but resistance has been reported [61,62].

There are several EOs that have shown activity against VREF (Appendix B, Table A2). Saki et al. investigated the antibacterial effects of cinnamon bark (*Cinnamomum zeylanicum*) EO on XDR isolates and determined that VREF was sensitive to this EO [63]. Five other EOs, namely bitter orange (*Citrus aurantium v. amara*), lemon (*Citrus × limon*), blue gum eucalyptus (*Eucalyptus globulus*), tea tree (*Melaleuca alternifolia*) and Mediterranean cypress (*Cupressus sempervirens*) were evaluated by Iseppi and collaborators. They observed that *M. alternifolia* EO was the most effective EO and *C. aurantium* the least effective EO, while the other three EOs displayed antibacterial activity against all strains, to a certain extent. What is more, EO–EO and EO–AB associations showed a synergistic antimicrobial activity in most tests and were even effective against biofilm formation [64]. Synergistic interactions were also investigated by Owen et al. in their study on oregano (*Origanum compactum*), rosewood (*Aniba roseadora*) and cumin (*Cuminum cymimum*) EOs. They found out that the combination of carvacrol and cuminaldehyde could re-establish susceptibility to vancomycin in VREF, resulting in bactericidal activity [65]. Their previous studies also indicated that these particular EOs exhibited zones of inhibition against VSEf and VREF in the Kirby–Bauer disc diffusion method, with minimal inhibitory concentrations (MICs) ranging from 0.29 to 37.20 mg/mL [66].

When looking into the effect of five EOs on VREF isolated from wastewater treatment plants, clinical samples and reference strains, Sakkas et al. discovered that origanum (*Thymus capitatus*), thyme (*Thymus vulgaris*) and tea tree (*Melaleuca alternifolia*) EOs were effective against the pathogenic strains [67]. Di Vito et al. compared the antimicrobial activity of EOs and hydrolates extracted from lavender (*Lavandula angustifolia* and *Lavandula intermedia*), origanum (*Origanum hirtum*), winter savory (*Satureja montana*), scarlet beebalm/bergamot (*Monarda didyma*) and beebalm/wild bergamot (*Monarda fistulosa*). They determined that the antimicrobial activity of hydrolates is milder than that of the corresponding EOs, with higher MICs. In contrast to hydrolates, which must be used at concentrations of 25–50% *v*/*v* to achieve the same antimicrobial activity, EOs are active at lower concentrations: 50% of the strains are susceptible at concentrations of 0.125–2% *v*/*v* [68].

Further data from these studies are presented in Appendix B, Table A2.

### 3.2. Helicobacter Pylori, Clarithromycin-Resistant

*Helicobacter pylori* are Gram-negative bacteria that colonize the stomach and duodenum, and might be a key contributor in diseases such as chronic gastritis, peptic ulcer, gastric adenocarcinoma, mucosa-associated lymphoid tissue lymphoma and iron deficiency anaemia [69]. While the most common sources of *H. pylori* contamination are water, environment and animals [70], it appears that a plant-based diet is associated with a lower prevalence in adults; in contrast, higher consumption of fried food and well water are considered risk factors for infection [71]. Overall, prevalence of *H. pylori* infection seems to be declining globally, with Oceania having the lowest prevalence rates (24.4%); however, in some parts of the world, the prevalence remains quite stable and reaches a rate of 79.1% in Africa [72]. *H. pylori* can cause lifelong infection without eradication, with recurrence appearing either by recrudescence or reinfection [73]. It has been established that *H. pylori* gastritis should be considered an infectious disease regardless of the absence of symptoms, complications or resultant illnesses [74]. Several specific virulence factors can lead to a more severe outcome: cytotoxin-associated gene A (*CagA*, especially EPIYA-D and EPIYA-C) and type IV secretion system (*CagL* polymorphism), the genotypes of vacuolating cytotoxin A (*vacA*, s1/i1/m1 types), and blood group antigen binding adhesin (BabA, low-producer or chimeric with BabB) [75].

The classical triple-therapy regimen against *H. pylori* infection includes a proton-pump inhibitor (PPI), amoxicillin and clarithromycin, taken together for 7–14 days [76]. The addition of nitroimidazole (metronidazole or tinidazole) adds benefits to the regimen, requiring fewer doses of ABs and being more effective [77]. Due to its ability to inhibit bacterial protein synthesis, clarithromycin has been the mainstay of therapy in *H. pylori* infections; the AB has a good mucosal diffusion, low MIC and a minimal impact on gastric acidity [78]. However, vertically transmitted point mutations in the peptidyl transferase loop of the V domain of 23S rRNA gene are leading to the emergence of CRHP strains [79]. In developed countries, 90% of CRHP appear as a result of mutations in two specific adjacent nucleotide positions—A2142G and A2143—which are the most prevalent and well-documented point mutations in this pathogen [80]. Efflux pumps and outer membrane proteins are also involved in clarithromycin-resistance. Notably, PPIs are structurally similar to efflux pump inhibitors, such as Phe-Arg-β-naphthylamide—a molecule able to decrease the MIC of ABs. As CRHP continues to escape efficient therapy, new drugs have been developed, such as vonoprazan—a novel potassium-competitive acid blocker that strongly inhibits H+/K+ ATPase-mediated gastric acid secretion [81]. In May 2022, The US Food and Drug Administration has approved two vonoprazan-based treatments, both superior to PPI-based triple or dual therapy [82].

Elkousy et al. demonstrated that both marjoram (*Origanum majorana*) and mandarin (*Citrus reticulata*) EOs exhibited antimicrobial activity against *H. pylori*. The authors stipulated that, thanks to its high content of oxygenated compounds (trans-sabinene hydrate, terpinen-4-ol, linalyl acetate, caryophyllene oxide and a-terpineol), marjoram EO displayed a lower MIC than the other EO. Additionally, the combination of the two EOs had a synergistic effect against *H. pylori*, with a lower MIC, equal to clarithromycin’s MIC [83] (Appendix B, Table A3). A study that investigated the effectiveness of four *Piper* spp. EOs against *H. pylori* determined that long pepper (*Piper longum*) EO recorded the same MIC as clarithromycin, followed by white pepper EO, tailed pepper and then black pepper. This study shows that, although the EOs came from plants in the same genus, individual components in each volatile oil led to variation in MIC values [84]. The same is true for different *Pinus* species, as Gad et al. determined that, among four pine EOs, *P. pinea* EO exhibited the highest anti-*H. pylori* activity, with a MIC comparable to that of clarithromycin [85]. In a study by Mariem et al., 54.54% of gastric biopsy *H. pylori* isolates showed resistance to at least one of the five tested ABs (erythromycin, clarithromycin, ciprofloxacin, levofloxacin or metronidazole), while mastic tree (*Pistacia lentiscus*) EO showed anti-*H. pylori* activity against all tested strains [86]. Several other EOs (wild thyme [87], cinnamon [88], cedarwood, oregano [89]) exhibited strong antimicrobial activity against *H. pylori* strains, while other EOs (common sage, lemon balm, English lavender [87], clove, thyme, rosemary [88], guabiraba [90]) showed good to mild antimicrobial effects. Various active compounds/EOs, such as geraniol [91], *α*-pinene (from *P. atlantica*) [92] or *β*-caryophyllene [93] are increasingly being tested on animal models, uncovering their anti-*H. pylori* activity (Appendix B, Table A3). However, data regarding EOs’ activity against CRHP are scarce.

### 3.3. Campylobacter spp., Fluoroquinolone-Resistant

*Campylobacter* spp. includes several commensal Gram-negative species, but also pathogenic strains of *Campylobacter jejuni* and *Campylobacter coli* [94]. *C. jejuni* alone is the leading cause of bacterial gastroenteritis in humans, surpassing *Escherichia coli*, *Shigella* spp. and *Salmonella* [95]. The main sources of contamination are raw or undercooked meat, unpasteurized milk or dairy products, contaminated vegetables, natural mineral water, shellfish and flies [96]. In developing countries, *Campylobacter* infection is generally limited to children and the clinical symptoms are usually indistinguishable from those caused by other enteric pathogens [97]. In some instances—usually in immunocompromised, pregnant and elderly patients—*Campylobacter* can trigger extraintestinal manifestations, including abscesses, meningitis, endocarditis and bacteraemia [98]. Moreover, a number of patients develop chronic sequelae, such as irritable bowel syndrome, reactive arthritis, Reiter’s Syndrome, Guillain–Barré Syndrome, Crohn’s disease and ulcerative colitis [99]. The main virulence factors of *Campylobacter* spp. are their ability to adhere and colonize (*cadF*, *racR*, *virB11*, *pldA* and *dnaJ* genes), invade intestinal epithelial cells (*ciaB* and *ceuE* genes) and produce toxins (*cdtA*, *cdtB*, *cdtC* genes) [100].

While self-limiting intestinal manifestations are treated using replacement of fluids and electrolytes, severe extraintestinal infections need treatment with the ABs of choice (macrolides and fluoroquinolones [101]) or with alternative solutions (tetracyclines and aminoglycosides), for which all have reported AMR [102]. Fluoroquinolones act by inhibiting DNA gyrase (GyrA and GyrB subunits) and topoisomerase IV (ParC and ParE subunits)—two enzymes crucial for bacterial survival, involved in DNA replication and repair, transcription and recombination [103]. Point mutations (T86I, T86K, A70T and D90N) in the quinolone-resistance-determining region (QRDR) of DNA gyrase’s GyrA subunit led to the emergence of FRC [104]. Moreover, drug efflux pump CmeABC works synergistically with the gyrA mutation, leading to the appearance of FRC; however, in the absence of the gyrA gene mutation, CmeABC over-expression does not correlate with ciprofloxacin resistance [105]. Nevertheless, fluoroquinolone resistance cannot be attributed to mutations in *parC* and *parE* genes (since Campylobacter lacks these particular genes); similarly, it appears that mutations in the *gyrB* gene are not related to the emergence of FRC [106].

Gahamanyi et al. evaluated the susceptibility of *C. jejuni* and *C. coli* to various natural products (plant extracts, EOs and active compounds) and frontline ABs. They determined that, among the tested products, cinnamon (*Cinnamomum cassia*) EO and its main compound (E)-Cinnamaldehyde, clove (*Syzygium aromaticum*) EO and its main compound eugenol, and baicalein had the lowest MIC and MBC values (25–100 µg/mL) [107] (Appendix B, Table A4). Duarte et al. tested the activity of coriander (*Coriandrum sativum*) EO and its major compound linalool against the two above-mentioned strains; once again both products showed comparable activities: inhibition of biofilm formation, reduction in quorum-sensing and inhibition of the microbial growth [108]. The anti-*C. jejuni* activity of clove EO and eugenol was supported by another study showing that the products perturb the expression of virulence factors, alter the morphology and induce oxidative stress in *C. jejuni* [109]. Ahmed et al. confirmed that clove and cinnamon EOs were effective against *C. jejuni* and *S. aureus*, but garlic EO did not share the same efficacy [110]. El Baaboua et al. determined that MDR *Campylobacter* spp. were sensitive to oregano (*Origanum compactum*), mint (*Mentha pulegium*) and lavender (*Lavandula stoechas*) EOs, which interfered with the microbial ability to form biofilms. Lavender and oregano acted synergistically with tetracycline or ampicillin, reducing the effective doses of EOs, tetracycline and ampicillin [111]. One interesting study investigated the potential of thyme (*Thymus vulgaris*) EO/gelatin nanofibers to inhibit *C. jejuni* growth in chicken. Food packaging containing thyme EO *β*-cyclodextrin *ε*-polylysine nanoparticles as antibacterial agents readily damaged *C. jejuni* cell membranes and reduced the microbial population [112]. Another noteworthy approach was that of Lin et al. who prepared chrysanthemum EO–chitosan–pectin triple-layer liposomes and determined that the product exhibited high anti-*C. jejuni* activity [113]. Lastly, even though some active compounds present promising in vitro results, further in vivo studies fail to reach the same outcomes. For instance, although (-)-*α*-pinene was able to reduce ciprofloxacin’s MICs (when used in combination), it did not manage to impede fluoroquinolone resistance development when added to enrofloxacin in broiler chickens [114] (Appendix B, Table A4).

### 3.4. Neisseria gonorrhoeae, Cephalosporin-Resistant, Fluoroquinolone-Resistant

*Neisseria gonorrhoeae* is a Gram-negative diplococcus and represents the etiological agent of gonorrhoea, the second most common sexually transmitted infection that affects both men and women [115]. According to WHO, *N. gonorrhoeae* causes significant morbidity and economic costs around the world, with 82.4 million new cases of gonorrhoea among adults and adolescents each year [116]. While the majority of gonorrhoea cases are asymptomatic, untreated infections may lead to serious complications such as endometritis, salpingitis, sterility, chronic pelvic pain, ectopic pregnancy, neonatal infections and increased risk of acquiring HIV [117]. Moreover, there are cases when the pathogen disseminates, causing skin, joint or tendon infection and, rarely, endocarditis or meningitis [118]. The primary step in *N. gonorrhoeae* pathogenesis is the bacterial adherence to the epithelium of the mucosa, mediated through surface structures: type IV pili, opacity (Opa) proteins, LOS and PorB—the major outer membrane protein porin [119]. After adhering, the pathogen replicates, forming microcolonies and biofilms, and sometimes even invades epithelial cells by transcytosis [120].

For many years, gonorrhoea was considered relatively easy to treat in monotherapy; however, due to AB overuse, AMR has emerged for most classes (sulphonamides, penicillins, tetracyclines, macrolides, fluoroquinolones and early-generation cephalosporins) [121]. Today, the most recommended gonorrhoea treatment is dual therapy: a single dose of a third-generation cephalosporin (250–500 mg intramuscular ceftriaxone or 400 mg peroral cefixime) in combination with azithromycin (1–2 g peroral) [122]. Concerningly, in 2018, ceftriaxone-resistant and azithromycin-resistant *N. gonorrhoeae* strains have been isolated [123]. Cephalosporin resistance is caused by mutations in various chromosomal regions that encode for important microbial proteins such as the transpeptidase domain of the PBP2 protein (*penA* gene), porin B subunit (*porB1b* gene) and PBP1 protein (*ponA* gene) [124], as well as overexpression of MtrCDE membrane pump proteins [125]. Fluoroquinolone resistance is acquired through mutations in QRDR: a single amino acid change in the GyrA subunit (positions 91 or 95) leads to intermediary resistance level, while three or more changes in the GyrA subunit (positions 91, 95 and 102), ParC (position 87 and 91) and/or Par E (position 439) proteins may lead to even higher MICs [126]. There is a pressing need to develop new diagnostic strategies and novel antimicrobials in order to preserve ceftriaxone, as it is the last empirical first-line monotherapy for gonorrhoea [127]. In 2021, WHO issued a general protocol called the Enhanced Gonococcal Antimicrobial Surveillance Programme, which aims to strengthen the quality, comparability and timeliness of gonococcal AMR data across multiple countries [128]. As the global burden rises, new collaborators join this programme to combat AMR in gonorrhoea [129].

Propolis extract has been shown to exhibit antimicrobial activity against ciprofloxacin-sensitive and ciprofloxacin-resistant *N. gonorrhoeae* strains, in a study by Vică et al.; depending on the harvesting region, the extracts presented with various inhibition zone diameters and MICs [130] (Appendix B, Table A5). Umaru et al. studied the effect of *Molineria capitulate* fruit EO on various pathogens and showed that both *M. capitulate* EO and its major component, myrcene, displayed antimicrobial activity against *N. gonorrhoeae* strains [131]. Soliman et al. investigated the antimicrobial properties of two guava (*Psidium* spp.) EOs and determined that both EOs showed good antibacterial effects, but *Psidium cattleianum* displayed preferential activity against *N. gonorrhoeae* [132]. Other publications showed that *Eclipta alba* EO is highly effective against *N. gonorrhoeae* strains [133], while *Ferula tingitana* EO displays a more modest antibacterial activity [134] (Appendix B, Table A5).

### 3.5. Salmonellae, Fluoroquinolone-Resistant

*Salmonella* spp. is a genus of facultative anaerobe Gram-negative bacilli having peritrichous flagella [135]. The genus *Salmonella* belongs to the family Enterobacteriaceae and includes two main species: *Salmonella enterica* and *Salmonella bongori* [136]. Around 99% of the *Salmonella* strains that cause infection in humans or other mammals belong to the *S. enterica* strains [137]. *S. enterica* is further subdivided into six-subspecies, serotypes, serogroups and serovars, according to the expression of somatic lipopolysaccharide O and flagellar H antigens [138]. The three major diseases caused by *Salmonella* in humans are: non-invasive non-typhoidal salmonellosis (niNTS), invasive non-typhoidal salmonellosis (iNTS) and typhoid fever [139]. Additionally, *Salmonella* serotypes can asymptomatically colonize humans’ gallbladders, thus making them chronic carriers and potential disseminators [136].

Non-typhoidal salmonellosis (NTS) is among the most prevalent global cause of foodborne illnesses [140] and includes infections caused by all *Salmonella* spp., with the exception of the distinct typhoidal serotypes: Typhi and Paratyphi A-C [139]. While niNTS may have a variety of clinical manifestations, the most common is gastroenteritis and is usually self-limiting [140]. Given that both humans and animals are potential hosts for niNTS [141], the infection is mainly transmitted via the consumption of animal products, but also unpasteurized dairy products, seafood and fruits [140]. The incidence of gastroenteritis due to niNTS peaks in the developing world, but it is also of considerable importance in developed countries; for instance, in the European Union (EU), salmonellosis is the second-most reported gastrointestinal infection in humans after campylobacteriosis [140]. When non-typhoidal *Salmonella* spp. (*S. typhimurium* and *S. enteritidis*) go beyond the gastrointestinal tract and invade normally sterile sites causing bacteraemia, iNTS occurs [142]. Contrary to niNTS, iNTS typically manifests as a febrile systemic illness (where diarrhoea is often absent) and lower respiratory tract disorders, due to co-infections with *Mycobacterium tuberculosis* and *Streptococcus pneumoniae* [139]. The global burden of iNTS is estimated by Ao et al., who evaluated 3.4 million cases of iNTS and approximated the annual death toll to around 700,000 [143]. In a similar study that excluded patients with HIV-associated iNTS, it was established that more than 60,000 deaths occurred in 600,000 cases of iNTS [142]. The most severe *Salmonella* spp. infections, typhoid and paratyphoid enteric fevers, are caused by *Salmonella enterica* subspecies *enterica* serovars Typhi and Paratyphi A, B and C [137,141]. Contrary to NTS broad host specificity, *S. typhi* is found only in humans [139]. For reasons not fully understood, it is estimated that around 5% of infected individuals will fail to clear the infection within a year, and will instead progress to a chronic carrier state where the bacteria will primarily reside in the hepatobiliary tract and gallbladder [139]. These systemic diseases cause more than 200,000 deaths globally, sub-Saharan Africa and Asia accounting for around 46% and 32% of typhoid fever cases, respectively [137]. Worryingly from 1990 to 2010 annual mortality from typhoid fever has increased by 39% [139].

Antibiotic therapy is not needed for *Salmonella*-induced gastroenteritis while, for invasive *Salmonella* infections, ampicillin, chloramphenicol and trimethoprim-sulfamethoxazole are used as first-line treatment [136]. However [139], a large proportion of AB-resistant *Salmonella* are acquired through the consumption of contaminated food of animal origin [144]. Poultry are the main source of human salmonellosis. In order to move forward through the food chain, *Salmonella* must be resistant to different environmental stress conditions such as heat, desiccation, nutrient starvation or biocides, and *Salmonella* spp. use quorum sensing to control processes such as luminescence, sporulation, virulence or biofilm formation [135].

AMR can be achieved by mutations in different chromosomal loci that are part of a core set of genes, such as genomic islands and through exogenous resistance genes carried by mobile genetic elements that can diffuse horizontally [145]. *Salmonella* spp. mediate resistance to ABs by three major mechanisms: drug inactivation, protection of the AB target sites, and removal of ABs using efflux pumps or multidrug pumps. Firstly, the main mechanism, drug inactivation, is characterized by destruction of antimicrobial agents (quinolones, macrolides) through chemical modification using enzymes that catalyse reactions such as acetylation, phosphorylation and adenylation [144]. Secondly, *Salmonella* spp. can protect the target sites of ABs, which are typically either enzymes or other specific cell structures [144,146]. For instance, the plasmid-encoded quinolone resistance protein (Qnr) confers resistance to quinolones by acting as a DNA homolog that competes for the binding of DNA gyrase and topoisomerase IV [144,146]. Thirdly, *Salmonella* spp. can use relatively nonspecific efflux pumps (such as AcrAB-TolC) encoded by genes within mobile elements to reject fluoroquinolones, β-lactams and carbapenems [144,146].

Emerging MDR *Salmonella* spp. have changed the treatment regimen towards using second-line ABs such as fluoroquinolones (ciprofloxacin) and third-generation cephalosporins [136]. However, as fluoroquinolones are extensively exploited for animal production in several countries [147], WHO notifies that the prevalence of FRS is growing quickly [136]. Under normal conditions, quinolones enter bacteria through porins and then exert their bacteriostatic activity by binding to the gyrase/topoisomerase IV–DNA complex [144]. Resistance to nalidixic acid, driven by a single mutation within *gyrA*, is a precursor to resistance to all quinolones [148]. Additionally, the fluoroquinolone resistance in *Salmonella* is caused either by chromosomal mutations in the QRDRs of the *gyr* and *par* genes [149], or by the acquisition of several plasmid-mediated quinolone resistance (PMQR) genes [143]: (1) *qnr *(quinolone resistance proteins); (2) *aac(6′)-lb-cr* (aminoglycoside-modifying acetyltransferase); and (3) *oqxAB* and *qepA* [141,144]. Increased expression of the AcrAB-TolC multidrug efflux system was also shown to play an important role in the development of high-level fluoroquinolone resistance [149]. Generally, resistance to ciprofloxacin in enteric bacteria is acquired through *gyrA* mutations, while PMQR genes also induce low-level resistance. In contrast, in *Salmonella* spp., mutations in transferable PMQR gene *qnr* were observed in all ciprofloxacin-resistant isolates, whereas *gyrA* mutations were often found in isolates with reduced ciprofloxacin susceptibility but not in all ciprofloxacin-resistant isolates [150].

To our knowledge, there are no articles describing the antimicrobial action of EOs on FRS.

### 3.6. Staphylococcus aureus, Methicillin-Resistant, Vancomycin-Intermediate and -Resistant

*Staphylococcus aureus* is a ubiquitous Gram-positive, coagulase-positive, facultative anaerobe, nonmotile and spherical bacterium (grape-like cluster), possessing not a flagella but a capsule [151]. It can be present either as a commensal member of the microbiota (usually localized in the upper respiratory tract, gut or on the skin) or as an opportunistic pathogen [152]. *S. aureus* may cause mild to severe infections, depending on the localization of the infection (skin, soft tissues or blood) and patient’s characteristics (age—more aggressive in infants and older people; comorbidities—immunosuppression, diabetes, heart or renal diseases; and others factors—implantable medical devices, low social economic status or intravenous drug use) [153,154]. The disorders caused by *S. aureus* range from skin infections (pimples, impetigo, folliculitis, boils, cellulitis, carbuncles, scalded skin syndrome, abscesses) to more severe and life-threatening diseases, including pneumonia, osteomyelitis, endocarditis, meningitis, toxic shock syndrome, bacteremia or sepsis [151,155].

While most people who are colonized with *S. aureus* will not develop an invasive infection, *S. aureus* infections are overall extremely frequent and particularly problematic due to AB resistance and the ability to form biofilms [153]. In 1942, shortly after the introduction of penicillin into clinical practice, *S. aureus* started hydrolysing the beta-lactam ring and establishing penicillin resistance. Then, in the late 1950s, the appearance of the semi-synthetic beta-lactamase-resistant AB, methicillin, led to the emergence of the first MRSA strain only 2 years after its introduction [156] due to mutations in the gene encoding for penicillin-binding protein 2a or 2′ (PBP2a; PBP2′) (*mec*A) [152]. Nowadays, *S. aureus* remains one of the most common resistant pathogens worldwide [154,157]. According to the American Centers for Disease Control and Prevention, *S. aureus* has evolved into a major health-threatening pathogen, as invasive infections caused by MRSA have a high mortality rate and its remarkable ability of acquiring AB resistance against multiple drug classes significantly complicates the treatment [158]. Although the rates of AB resistance are widely variable depending on the country (relatively low AMR in Scandinavian countries and high AMR in Southern Europe, USA and China, due to differences in hygiene and surveillance measures), they are still on the rise in poorly developed countries (South America and some countries in Africa) [151,159].

The beginning of the 1960s was marked by the discovery of the SCC*mec* complex, a key factor that enabled *S. aureus* to acquire resistance to most of the beta-lactam ABs [151]. Currently 12 SCC*mec* complexes are known, and they are divided by (1) the group of cassette chromosome recombinase (ccr) complex and (2) the category of *mec* complex. With the exception of type XI SCC*mec* which contains homologue *mec*C, all SCC*mec* types include *mec*A, a component that encodes for PBP2a [160]. PBP2a is responsible for the transpeptidase action in the biosynthesis of peptidoglycan, in the presence of beta-lactam AB, inhibiting the function of PBP 1, 2, 3 and 4. While *mec*C is a variant of *mec*A which encodes for PBP2aLGA251 (named after MRSA strain LGA 251), *mec*B is a plasmid-developed methicillin resistant form with unclear mechanism of resistance [161,162]. *mec*A’s expression depends on regulators encoded by *mec*I*, mec*R1 and *mec*R2, and on regulators of gene expression, such as *bla*Z, *bla*I and *bla*RI [162]. The auxiliary *fem* genes seem to also have an important influence on the resistance phenotypes [163].

Over time *S. aureus* has also gained resistance to vancomycin, the AB used as a first-line treatment for the past six decades [164,165]. The feasible alternative options to vancomycin treatment include high doses of daptomycin co-administered with either gentamicin, rifampicin, linezolid, trimethoprim + sulfamethoxazole or beta-lactam. If response to daptomycin is inadequate, secondary options include monotherapy or co-administration with a number of possible ABs: quinupristin + dalfopristin, linezolid, telavancin or trimethoprim + sulfamethoxazole [166]. In 2002, VRSA strains were identified in the USA; it is thought that resistance was mediated by *van*A gene acquired from *E. faecalis* on the plasmid-borne transposon Tn1546 [167]. Given that VRSA has a preference for diabetic wounds where vancomycin-resistant enterococci reside, there is a clear opportunity for horizontal gene transfer of Tn1546 accommodating *van*A [168].

The study performed by Oo T et al. evaluated the antimicrobial efficacy of crude extract and EO obtained from nutmeg (*Myristica fragrans* Houtt.) on *S. aureus* efflux pump systems (chromosomal norA and mepA) involved in the resistance mechanism of MRSA. They found that elemicin, myristicin, methoxyeugenol and asarone can work as efflux pump inhibitors, thus potentiating the antimicrobial activity of classical ABs. They observed a synergistic activity of ciprofloxacin and the two nutmeg formulations, concluding that both the extract and the EO act as efflux pump inhibitors, while ciprofloxacin acts as an efflux system [169] (Appendix B, Table A6). Alharbi NS et al. highlighted the effect of two different concentrations of tailed pepper (*Piper cubeba* L.) EO against MRSA ATCC 43300. While the higher concentration induced serious microscopic deteriorations of the bacterial cell, the lower concentration had no observable microscopic effects; however, significant modifications within the cell wall were observed at a nanoscopic level. The authors concluded that the EO induced an antibacterial action on both methicillin- and oxacillin-resistant *S. aureus* strains through its action upon the cell wall and the cytoplasmic membrane [170] (Appendix B, Table A6).

Piasecki B et al. tested 19 EOs extracted from *Cymbopogon* spp. and determined that *C. flexuosus* (lemongrass) EO exhibited the highest antibacterial activity, while citronellol stood out as the most powerful active compound (from citronellol, geraniol and citral). Moreover, all tested EOs manifested antibiofilm properties, with a MBIC ranging from 1 to 4 mg/mL. Nonetheless, after 48 h of treatment at a maximum concentration, cardiotoxicity and shortened tail were observed in zebrafish, with *C. martini* var. *motia* showing the most toxic potential (about 20 times more toxic than *C. winterianus*) [171]. Merghni A et al. also showed that both blue gum (*Eucalyptus globulus* Labill.) EO and its main active compound, 1,8-cineole, present excellent antibiofilm properties and bacteriostatic effects, with the EO inducing a more potent effect on quorum sensing [172] (Appendix B, Table A6).

It is well known that Gram-negative bacteria are more resistant to ABs and toxins than Gram-positive bacteria, due to their perfected cell wall. Specifically, they have a multi-layered, complex cell wall, covered by a hydrophilic membrane abundant in lipopolysaccharides. Moreover, the periplasmic space of Gram-negative bacteria contains enzymes capable of degrading exogenous molecules, thus preventing the access of inhibitors. Several studies have also emphasized that there is a difference in the antibacterial activity of EOs in Gram-positive versus Gram-negative bacteria [173] (Appendix B, Table A6).

Predoi D et al. investigated the activity of EOs in hydroxyapatite, a calcium phosphate compound best known for its similarities with human hard tissues and its applicability in medicine (dental applications, bone regeneration). Their work was based on the premise that, by incorporating antibacterial agents in hydroxyapatite, the risks of postoperative infections would be reduced. Thus, they highlight the physio-chemical properties and antimicrobial activity of two nanocomposites of hydroxyapatite embedded with basil and lavender EOs, with the latter exhibiting the best antibacterial properties [174]. The team continued to look into the effectiveness of hydroxyapatite associations with EOs in further studies [175,176] (Appendix B, Table A6).

Interestingly, Mouwakeh A and collaborators postulated the fact that black caraway (*Nigella sativa* L.) EO and its active components (carvacrol and p-cymene) could be used as MRSA modifiers of resistance. In their study they hypothesized that the hydroxyl group in carvacrol and thymoquinone might play a key role in their antimicrobial activity [177].

De Moura et al. focused on evaluating the antioxidant, antibacterial and antibiofilm activity of nerolidol, an acyclic sesquiterpene present in many species (usually the trans isomer) such as wood oil, red oil, cabreuva oil and balm of Peru [178]. It is of valuable importance in the cosmetic industry (as a preservative agent to fixate perfumes) and pharmaceutical industry (as a stimulating agent to increase the concentration of active ingredients in transdermal formulations) [179]. Nerolidol has a broad spectrum of pharmacological/biological actions, including antineoplastic, anti-inflammatory, antinociceptive, larvicidal and leishmanicidal activity [179], and, recently, its implication for neurodegenerative disorders has been described [180]. De Moura and colleagues found that nerolidol is effective for both MSSA and MRSA at the same MIC of 2 mg/mL [178].

Two EOs of lemon verbena (*Aloysia citriodora* Palau) collected from different Palestinian regions were tested for their antimicrobial, antioxidant, cytotoxic and cyclooxygenase (COX) inhibitory effects by Jaradat N et al. Both EOs showed antimicrobial activity against MRSA, *P. vulgaris* and *C. albicans*, but the Baqa al-Gharbiyye EO manifested stronger antioxidant, cytotoxic and anti-cyclooxygenase activities, compared with the Umm al-Fahm EO [181] (Appendix B, Table A6).

The work of Adrian Man and collaborators investigated the antimicrobial effect of some well-known EOs in micellar and aqueous extracts. They assessed the antibacterial activity of oregano, lemon, thyme, myrtle and frankincense EOs against *S. aureus*, *E. faecalis*, *E. coli*, *K. pneumoniae* and *P. aeruginosa*, and determined that gram-positive bacteria (including MRSA) were more susceptible compared with *P. aeruginosa* which was found to be the most resistant. The authors highlighted that micellar suspensions of EOs (especially those containing high concentrations of terpenes and terpenoids—i.e., oregano, thyme, lemon EOs) can be introduced in new topical formulations to enhance the penetrability of EOs and, thus, their action [41] (Appendix B, Table A6).

The study of Kwiatkowski P. addressed the efficacy of several active compounds in EOs against mupirocin-susceptible and low-level mupirocin-resistant MRSA. Notably, mupirocin is an AB synthesized by *Pseudomonas fluorescens* and has medical applications. It is used on nasal mucosa as an ointment in order to decolonize *S. aureus*. It is applied for 5–14 days (especially before a surgical intervention) to reduce the risk of postoperative wound infection and prevent the spread of bacteria to medical staff’s hands [182]. Screening of *S. aureus* is mandatory in patients undergoing cardiac or orthopaedic surgery, especially for MRSA. Moreover, all patients prior to surgical procedures must have a full-body shower (with/without disinfectants such as chlorhexidine) to reduce the number of surgical site infections and complications caused by this highly pathogenic and resistant microorganism [183]. The mechanism by which mupirocin exerts its antibacterial activity consists of the inhibition of isoleucyl-tRNA (ended by ileS gene on the chromosome) and, thus, the blocking of protein synthesis within the bacteria. Two types of mupirocin resistance have been described: low-level (MupRL, MIC: 8–256 mg/L) and high-level (MupHR, MIC: 512 mg/L). Usually, the MIC of mupirocin on sensible strains is around 4 mg/L. If MupRL is caused by ileS gene mutation, MupRH is induced by a plasmid encoded in the ileS2 gene (responsible for encoding a different isoleucyl-tRNA synthase) which will trigger a low affinity for mupirocin [182]. In the study performed, carvacrol highlighted the best inhibitory action on the tested MRSA strains and 1,8-cineole induced a synergistic action against MupRL MRSA with penicillin G. The authors suggested that high precision technology that trigger crucial checkpoints for staphylococcal resistance is needed [184] (Appendix B, Table A6).

The group of Manzuoerh et al. investigated the in vivo effect of topical dill (*Anethum graveolens*) EO versus mupirocin on a MRSA-induced infection in a BALB/c mouse model. They used two different concentrations of dill EO and determined that the main active components were α-phellandrene (47.3%), *p*-cymene (18.5%) and carvone (14.1%). The topical administration of dill EO decreased the inflammation and stimulated re-epithelialization, angiogenesis and collagen and fibroblast sedimentation. The topical effects were the result of an increased expression of p53 and caspases-3, in the case of the anti-inflammatory activity, and Blc-2, VEGF and FGF-2 expression in the case of the proliferative activity. Together with the stimulation of collagen synthesis through increases of ERα expression level, all these effects led to improvements in wound healing and reduction of the infection [185]. Mahboubi M et al. performed a similar study on EOs obtained from the aerial parts of *Oliveria decumbens* and *Pelargonium graveolens*. The main active substances in *Oliveria decumbens* EO were thymol (50.1%), *γ* -terpinene (20.7%) and p-cymene (17.6%), while beta-citronellol (39.3%) and geraniol (23.6%) were present in *Pelargonium graveolens* EO. Both the herbal cream containing the two EOs and the mupirocin formulation diminished log CFU (colony-forming units). Although further clinical and toxicological studies are required, topical formulations of EOs could be used in the treatment of wound infections, given their antimicrobial activity and healing effects [186] (Appendix B, Table A6).

In 2020, Chen J and collaborators presented the metabolomics analysis of antibacterial activity of camphor leaves EO (*Cinnamomum camphora*) [187]. It is worth mentioning that metabolomics analysis contributes to a multifactorial description of a drug mechanism of action and identifies eventual dynamic changes in metabolites as a response to drug treatment [188]. Moreover, key marker identification through pathway analysis can be used in order to identify metabolites’ changes in the course of antimicrobial activity. The camphor EO was rich in linalool (26.6%) and eucalyptol (16.8%), as well as α-terpineol (8.7%), isoborneol (8.1%), *β*-phellandrene (5.1%) and camphor (5.0%). The metabolomics analysis revealed 74 different metabolites (29 upregulated and 45 downregulated). The EO stimulated the activity of isocitrate dehydrogenase (47.35%) and decreased the activity of malate dehydrogenase (72.63%), succinate dehydrogenase (31.52%) and α-ketoglutarate dehydrogenase (63.29%). The authors concluded that the antimicrobial activity was the result of an imbalance in the amino acid metabolism, a rise in the apoptosis rate and a disruption in cell wall and membrane with the efflux of DNA, RNA and proteins from the MRSA cell [187] (Appendix B, Table A6).

There is a scarcity of studies describing the antimicrobial properties of EOs against VRSA and VISA strains. Vasconcelos SECB et al. investigated the antibacterial activity of Mexican mint (*Plectranthus amboinicus*) EO on VRSA and oxacillin-resistant *S. aureus* and determined that the strains were more sensitive when both carvacrol and EO were used [189] (Appendix B, Table A6).

Further data from these and other studies are provided in Appendix B, Table A6.

## 4. Conclusions

This systematic review presents the most recent studies of EOs’ activity against pathogens on WHO’s priority 2 list. While the emergence of AMR is a natural evolutionary process in bacteria, the widespread use and misuse of ABs has had an amplifying effect on this process.

Due to their selectivity on pathogenic bacteria and relatively low toxicity, EOs have been proposed as good alternatives and valuable adjuvants in a variety of infections. However, testing and evaluating the antimicrobial activity of EOs is difficult because of their volatility, water insolubility and complexity. Moreover, long incubation times during the testing period may result in the evaporation or decomposition of some of the active components. EOs’ phytocompound composition can vary and is influenced by plant subspecies, geographical location, growing conditions, growth phase, extraction method, and exposure to light, temperature and humidity.

There is a need for a consensus on the methodology used to assess the antimicrobial activity of EOs. In our paper, we refrained from a strict assessment of the results in the reviewed studies, given the heterogeneity of research design and techniques. Further research incorporating standardization methods to form a universal consensus are needed to allow a critical examination of EOs’ antimicrobial action. As a general consideration, some authors interpret the findings as follows: MIC: <100 µg/mL—highly active; MIC = 100–500 µg/mL—active; MIC = 500–1000 µg/mL—moderately active; MIC = 1–2 mg/mL—low activity; and MIC: >2 mg/mL—inactive [141,149].

Interestingly, suggestions that might enable the use of EOs in therapy have been mentioned as limitations in some of studies cited above. Thus, in order to provide cohesive results and to facilitate the use of EOs in clinical practice, the following should be considered: (1) assurance that the techniques used to analyse EOs’ composition are compliant with pharmacopeial requirements; (2) investigation of at least 10 different strains of the same microbial species in the same study, as they vary greatly from one another; (3) inclusion of anti-biofilm activity; (4) implementation of cytotoxicity assays for each study as there are limited data regarding EO dosage and safety, especially in humans; (5) increased standardization in methodology; and (6) intensive monitoring over time. Until further research elicits sufficient data for the large-scale use of EO phytocompounds in the treatment of infection, the general population might incorporate EOs into their daily diet for prophylaxis. For instance, not all cultures take the best benefits from using spices, even though they have good, non-specific and multidirectional antimicrobial activity.

In conclusion, while there is growing evidence for the introduction of EOs into clinical practice, especially those which observed no toxic effects, they are currently underused in practice. Based on their efficacy from the combined action of antimicrobial compounds and their synergistic activity with conventional ABs or food preservatives, EOs have the potential to lessen the burden of AMR. The development of standardized techniques for analysing the in vivo antimicrobial activity of EOs is a critical first step.

## Figures and Tables

**Figure 1 ijms-24-09727-f001:**
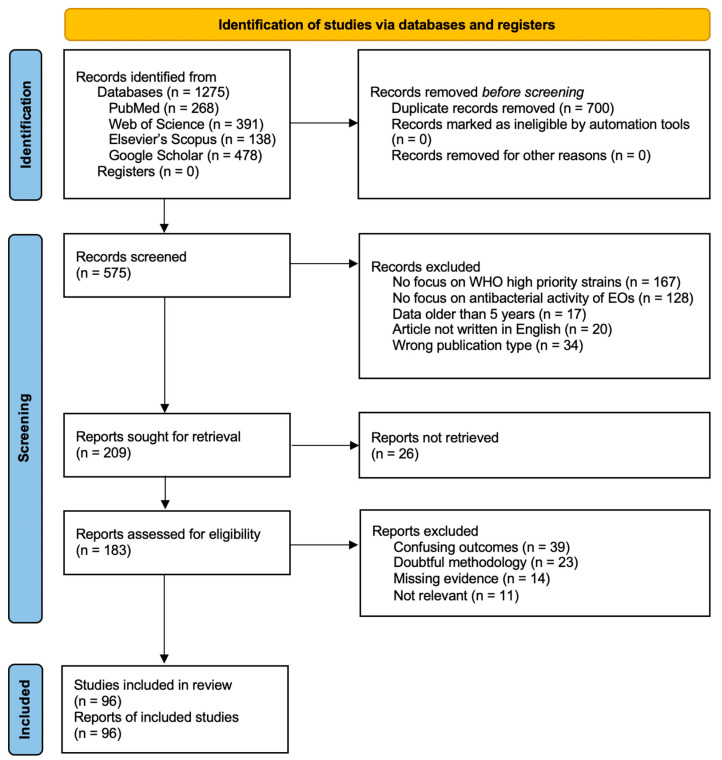
PRISMA 2020 flow diagram for new systematic reviews which included searches of databases and registers only: following database identification, duplicates were removed and the remaining records were screened for meeting the inclusion criteria. Retrieval of the full text was not possible for all publications. Before extracting the data from the studies, a final assessment was carried out to eliminate publications whose findings were ambiguous, according to the exclusion criteria.

**Table 1 ijms-24-09727-t001:** Top 5 countries in terms of AMR to HPPs, according to One Health Trust’s Resistance Map [31].

Bacteria	AB Resistance	Year	Resistant Strains Proportion (%)	Country
*E. faecium*	Vancomycin	2017	69 (62–75)	Argentina
		2016	68 (65–71)	USA
		2016	66 (60–71)	Taiwan
		2019	60 (52–67)	Serbia
		2012	51 (42–60)	Venezuela
*S. aureus*	Methicillin	2019	88 (77–95)	Egypt
		2017	73 (69–77)	Vietnam
		2019	68 (66–69)	India
		2018	66 (60–72)	Nigeria
		2019	65 (59–70)	Pakistan

## Data Availability

All the extracted data is presented within the manuscript.

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
