# Peer review of "Current State of Knowledge Regarding WHO High Priority Pathogens—Resistance Mechanisms and Proposed Solutions through Candidates Such as Essential Oils: A Systematic Review"

_ijms, 2023, doi:10.3390/ijms24119727_

Round 1

Reviewer 1 Report

The review is well organized and comprehensive; although it is too lengthy. However, this review was published by the same authors in 2022, under the title of: "Current state of knowledge regarding WHO critical priority pathogens: Mechanisms of resistance and proposed solutions through candidates such as essential oils". Plants 11(4).. https://doi.org/10.3390/plants11141789.

The question that arises is: What is the novelty in the current review?

Author Response

Dear Reviewer 1,

Thank you very much for the time allocated for reviewing the MS- "Current State of Knowledge Regarding WHO High Priority Pathogens – Resistance Mechanisms and Proposed Solutions through Candidates such as Essential Oils: A systematic review "  as well as for your pertinent observations.

Please see below our response to your comment:

“The review is well organized and comprehensive; although it is too lengthy. However, this review was published by the same authors in 2022, under the title of: "Current state of knowledge regarding WHO critical priority pathogens: Mechanisms of resistance and proposed solutions through candidates such as essential oils". Plants 11(4).. https://doi.org/10.3390/plants11141789.

The question that arises is: What is the novelty in the current review?”

Response

                        We thank the reviewer for their comment and suggestions.

We would like to clarify that this current review is addressing the WHO high priority pathogens (priority 2), whereas our previous work tackled the critical priority ones (priority 1). This implies that we have discussed the activity of essential oils on a different set of bacteria (e.g., E. faecium, H. pylori, Campylobacter spp., N. gonorrhoeae, and S. aureus).

We concur that the length of our manuscript might affect the reader’s interest in the topic. We believe that the introduction has now been appropriately amended to only include relevant information. We have reduced the size of the Results and Discussion section in the manuscript and shifted the focus from the mechanisms of antimicrobial resistance to the antibacterial activity of essential oils.

Sincerely,

Cl. pharm. Valentina Buda
Associate Professor
Discipline of Clinical pharmacy, Communication in pharmacy and Pharmaceutical care

Faculty of Pharmacy
"Victor Babes" University of Medicine and Pharmacy, Timisoara, Romania

Reviewer 2 Report

The manuscript "Current State of Knowledge Regarding WHO High Priority Pathogens: Resistance Mechanisms and Proposed Solutions through Candidates such as Essential Oils" addresses an important issue: the growing antimicrobial resistance and how to combat it with essential oils.

The objectives were clearly stated and explained in the manuscript, however the experimental strategy raises some major concerns and so the experimental information from which the conclusions were drawn. The manuscript is overall well written and has good organization. Quantitative analysis of the experimental data is missing throughout the manuscript and the interpretations of the results and the discussion are thus suffering from these limitations.

The paper is interesting but there is a need for more experimental detail in order to critically review the data. Specifically, they should provide information for the following questions and comments:

Major points:

1.      The authors should include more recent update on this topic and compare how this study further advances the current knowledge in the “Introduction section”.

2.      Caption in Figure 1 is scarce, a more detailed description is needed.

3.      Unify the style of the references in the References Section and add DOI in the cases it is possible. And use the same reference and citation (follow MDPI’s guidelines) style in the main text.

4.      The Methods section in the study should be more accurately described for each technique used.

5.      Table 6 is immense, please consider moving it to the Supplementary Material Section.

Minor points:

1.      The amount of references is overwhelming, please check if all of them are necessary.

2.      The terms in vitro and in vivo should be italicized every time they are used.

I would like to comment on the quality of English language used in the manuscript. While the content of the manuscript appears to be well-researched and informative, the English language used in the manuscript needs some improvement.

I recommend that the authors work with a professional editor or native English speaker to improve the language used in the manuscript. This will help to ensure that the manuscript is clear, concise, and easy to understand for readers.

I believe that improving the quality of English language used in the manuscript will significantly enhance the readability and impact of the research presented.

Author Response

Dear Reviewer 2,

Thank you very much for the time allocated for reviewing the MS- "Current State of Knowledge Regarding WHO High Priority Pathogens – Resistance Mechanisms and Proposed Solutions through Candidates such as Essential Oils: A systematic review "  as well as for your pertinent observations.

Please see below our response to your comments:

“The manuscript "Current State of Knowledge Regarding WHO High Priority Pathogens: Resistance Mechanisms and Proposed Solutions through Candidates such as Essential Oils" addresses an important issue: the growing antimicrobial resistance and how to combat it with essential oils.

The objectives were clearly stated and explained in the manuscript, however the experimental strategy raises some major concerns and so the experimental information from which the conclusions were drawn. The manuscript is overall well written and has good organization. Quantitative analysis of the experimental data is missing throughout the manuscript and the interpretations of the results and the discussion are thus suffering from these limitations.

The paper is interesting but there is a need for more experimental detail in order to critically review the data. Specifically, they should provide information for the following questions and comments:

Major points:

  1. The authors should include more recent update on this topic and compare how this study further advances the current knowledge in the “Introduction section”.

  1. Caption in Figure 1 is scarce, a more detailed description is needed.

  1. Unify the style of the references in the References Section and add DOI in the cases it is possible. And use the same reference and citation (follow MDPI’s guidelines) style in the main text.

  1. The Methods section in the study should be more accurately described for each technique used.

  1. Table 6 is immense, please consider moving it to the Supplementary Material Section.

Minor points:

  1. The amount of references is overwhelming, please check if all of them are necessary.

  1. The terms in vitro and in vivo should be italicized every time they are used.

I would like to comment on the quality of English language used in the manuscript. While the content of the manuscript appears to be well-researched and informative, the English language used in the manuscript needs some improvement.

I recommend that the authors work with a professional editor or native English speaker to improve the language used in the manuscript. This will help to ensure that the manuscript is clear, concise, and easy to understand for readers.

I believe that improving the quality of English language used in the manuscript will significantly enhance the readability and impact of the research presented.”

Response

We thank the reviewer for their comments and have made significant revisions to the manuscript to address their concerns.

We understand the points the reviewer raised regarding the quantitative analysis of the experimental data. There is still disagreement over how to interpret the results of the antimicrobial activity of essential oils, as the values of MIC vary within a large interval. It is for this reason that we decided to conduct a systematic review, rather than a meta-analysis. To clarify, we based our conclusions on study limitations and suggestions of the authors we cited.

                        We have addressed the major points raised as follows:

  1. To better clarify the value of our study, we have updated the Introduction of the manuscript (please also check the last paragraph – objectives). Here, we introduced few additional references to support the antimicrobial utility of essential oils and their potential to be introduced into clinical practice. Of note, our most recent literature search did not result in substantial new results, compared to our initial investigation. For this, we are confident that we managed to cover at least most of the latest advances in the field. In the eventuality we failed to meet the reviewer’s expectations, we kindly ask for point clarifications, to be able to address them correctly.
  2. We have amended both the caption and the text of Figure 1, to improve its clarity.
  3. We revised the reference style and updated it accordingly.
  4. We have described (in depth) the main methods used to assess the antimicrobial activity.
  5. We moved all the tables to Supplementary material, Appendix B. We had 5 peer-reviewers for this article and some suggestions were difficult to be addressed and uniformized for all of them, in the same manuscript. One of the reviewers suggested to make just one table with the most important findings over a one-page length. We tried to do this by choosing the studies with the best MIC in terms of antimicrobial effect, but we found more than 25 studies through merging all the tables (which would have required more than one page length). Moreover, as we already mentioned in the manuscript, due to the the wide diversity of bacterial strains, types of extracts, methodological dissimilarities and study design variations, we finally concluded that the best thing to do is to keep all the tables in the initial format and to move all of them in the Supplementary data part.

The minor points were covered in the following manner:

  1. Given that we reduced the size of the introductory part for each high priority pathogen, we also cut down on the number of references. However, we believe that further reducing the content of the “resistance mechanism” part will have a negative impact on comprehending the pathogenicity of the bacterial strains.
  2. According to MDPI’s style guide (second edition), we avoided the use of italics for Latin terms that have been commonly used in English for a long time, such as “in vivo” or “in vitro”.

We agree that the English of our manuscript needed improvements and we have addressed this issue with a specialist.

Sincerely,

Cl. pharm. Valentina Buda
Associate Professor
Discipline of Clinical pharmacy, Communication in pharmacy and Pharmaceutical care

Faculty of Pharmacy
"Victor Babes" University of Medicine and Pharmacy, Timisoara, Romania

Reviewer 3 Report

The Romanescu team shows a manuscript focussing on the current state-of-the-art regarding High Priority Pathogens according to the WHO classification. They highlight the resistance mechanisms of these pathogens and the efficacy of essential oils as alternatives to conventional antimicrobial therapies. 

The article is a continuation of a previously published one which focused on Critical Priority Pathogens according to the WHO classification.

However, the present manuscript is too exhaustive, with a lot of irrelevant information important to the primary discussion which should be the efficiency of essential oils. Moreover, the authors divide the section according to microorganism but then tend to not be very successful in this separation (for example, in lines 304 to 334, in which the discussion is not only about VREF, or in lines 459 to 461, or in lines 526 to 538, among others).

Thus, in the eyes of this reviewer, the article should be reformulated to exclude all irrelevant information that is not directly related to the essential oils topic and a clear division between microorganisms showed be accomplished.

Furthermore, all tables reporting the full list of articles are very visually exhaustive and should be reallocated to supplementary information, with possibly the creation of a resume table with no more than one page to try to highlight all the important information regarding essential oils and their effect on microorganisms. 

Additionally, some minor comments and suggestions are also given:

Line 44 - Campylobacter should be italicized

Line 46 - Consider maintaining only methicillin-resistant Staphylococcus aureus and vancomycin-resistant Staphylococcus aureus

Line 78/79 - missing a reference

Line 80/84 - missing references

Line 107/108 - missing a reference

Line 120/121 - missing references

Line 144/146 - missing references

Line 154 - the first time a bacterial species is mentioned, the full scientific name should be stated (i.e. Pseudomonas aeruginosa instead of P. aeruginosa, for example). Rectify throughout the manuscript.

Line 159/163 - missing references

Table 1 - Why do the authors decide to highlight these specific countries from Europe?

Line 159/166 + Table 1 - Why did the authors decide to specifically highlight the European situation in this section, if the remaining article is not focused on Europe? Please consider removing this section.

Line 183 - remove "that"

Line 190 - Please rectify the chemical formula of carbon dioxide.

Figure 1 - It is not clear what the exclusion criteria applied in "records excluded" and in "reports not retrieved". It is also not clear the transition from 183 articles to 96 articles, since 183 - 23 - 14 - 11 = 135 and not 96. Please clarify.

Line 270 - replace "flora" with "microbiota".

Line 295 - The remaining van genes are located where?

Line 896/904 - Authors lose an incredible opportunity to try to collect and analyze these data by meta-analysis to try to provide recommendations to a standardized scale.

Author Response

Dear Reviewer 3,

Thank you very much for the time allocated for reviewing the MS- "Current State of Knowledge Regarding WHO High Priority Pathogens – Resistance Mechanisms and Proposed Solutions through Candidates such as Essential Oils: A systematic review "  as well as for your pertinent observations.

Please see below our response to your comments:

“The Romanescu team shows a manuscript focusing on the current state-of-the-art regarding High Priority Pathogens according to the WHO classification. They highlight the resistance mechanisms of these pathogens and the efficacy of essential oils as alternatives to conventional antimicrobial therapies.

The article is a continuation of a previously published one which focused on Critical Priority Pathogens according to the WHO classification.

However, the present manuscript is too exhaustive, with a lot of irrelevant information important to the primary discussion which should be the efficiency of essential oils. Moreover, the authors divide the section according to microorganism but then tend to not be very successful in this separation (for example, in lines 304 to 334, in which the discussion is not only about VREF, or in lines 459 to 461, or in lines 526 to 538, among others).

Thus, in the eyes of this reviewer, the article should be reformulated to exclude all irrelevant information that is not directly related to the essential oils topic and a clear division between microorganisms showed be accomplished.

Furthermore, all tables reporting the full list of articles are very visually exhaustive and should be reallocated to supplementary information, with possibly the creation of a resume table with no more than one page to try to highlight all the important information regarding essential oils and their effect on microorganisms.

Additionally, some minor comments and suggestions are also given:

Line 44 - Campylobacter should be italicized

Line 46 - Consider maintaining only methicillin-resistant Staphylococcus aureus and vancomycin-resistant Staphylococcus aureus

Line 78/79 - missing a reference

Line 80/84 - missing references

Line 107/108 - missing a reference

Line 120/121 - missing references

Line 144/146 - missing references

Line 154 - the first time a bacterial species is mentioned, the full scientific name should be stated (i.e. Pseudomonas aeruginosa instead of P. aeruginosa, for example). Rectify throughout the manuscript.

Line 159/163 - missing references

Table 1 - Why do the authors decide to highlight these specific countries from Europe?

Line 159/166 + Table 1 - Why did the authors decide to specifically highlight the European situation in this section, if the remaining article is not focused on Europe? Please consider removing this section.

Line 183 - remove "that"

Line 190 - Please rectify the chemical formula of carbon dioxide.

Figure 1 - It is not clear what the exclusion criteria applied in "records excluded" and in "reports not retrieved". It is also not clear the transition from 183 articles to 96 articles, since 183 - 23 - 14 - 11 = 135 and not 96. Please clarify.

Line 270 - replace "flora" with "microbiota".

Line 295 - The remaining van genes are located where?

Line 896/904 - Authors lose an incredible opportunity to try to collect and analyze these data by meta-analysis to try to provide recommendations to a standardized scale.”

Response

We appreciate the points the reviewer raised and adjusted the manuscript to address the concerns.

We amended the manuscript to provide a clear division between microorganisms and to avoid misinterpretations. Of note, the confusion might arise from the fact that some publications tackle multiple high priority pathogen at once; it is for this reason we occasionally treated more than one bacterium, when discussing particular studies.

The “resistance mechanism” part for each high priority pathogen has been revised and considerably reduced. However, we believe that further condensing this section will be detrimental to understanding the pathogenicity of the mentioned microorganisms.

Based on the wide diversity of bacterial strains, types of extracts, methodological dissimilarities and study design variations and in order that the reader have the complete set of data that were extracted from the included studies we decided to move all the tables to supplementary material (without any selection of the data).

The minor suggestions were tackled as follows:

  1. Figure 1 was revised to clarify all the missing information. We have also improved the figure’s caption to provide more details.
  2. Based on the frequency of resistant bacterial species, the countries in table 1 were arranged in descending order. Thus, we chose to show the first 5 countries (for each bacterial strain) for which the antimicrobial resistance was the highest.
  3. We amended the Introduction with a short section (and table) regarding the global situation of antimicrobial resistance, with the expected emphasis on high priority pathogens. Given that the global database is less comprehensive (probably due to insufficient reports), we chose to keep the information gathered from the European Atlas, but to provide the table as a supplementary material.
  4. The full scientific name was used when a bacterial species was mentioned for the first time.
  5. Italics were used for genus and species throughout the manuscript.
  6. Wherever references were missing, they were inserted.
  7. We believe that data from further studies are needed in order to be able to provide recommendations for a standardized scale.

Sincerely,

Cl. pharm. Valentina Buda
Associate Professor
Discipline of Clinical pharmacy, Communication in pharmacy and Pharmaceutical care

Faculty of Pharmacy
"Victor Babes" University of Medicine and Pharmacy, Timisoara, Romania

Reviewer 4 Report

Dear Editor and Authors,

AMR is a global health problem that WHO will address in 2021. Effective prophylactic measures and more effective and non-toxic antimicrobials are needed. Essential oils (EOs) are still newcomers when it comes to treating infections in the clinical setting, as there is insufficient data on their in vivo activity and toxicity. In this review, the authors address the concept of antibiotic resistance and its major determinants, the methodology used to address the problem, and the current state of research on the pathogenesis, mechanism of resistance and activity of various essential oils against six priority pathogens. The article is comprehensive, well written and very enlightening. However, in some parts the article seems to be tailored to the problem in Europe, especially Eastern Europe. Given the importance of the topic, it would be ideal if the authors made an effort to problematise the issue globally. Please, verify the specifics comments bellow.

L44 – Campylobacter (in italic);

L263-264 - Is there a problem with mentioning the name of the microbiologist in this case?

Fig. 1 - Congratulations on this figure. Very instructive for applied methodology.

L269 - I think that in these cases the scientific name should be emphasised in some other way. If it is a rule that subtitles are italicised, the scientific name should not be italicised. L337 (idem).

L338 - For stylistic reasons, it is advisable not to begin a sentence with the abbreviation of a name, even in the case of generic names. In this case, rather write the generic name without the abbreviation.

L479 - I think that in these cases the scientific name should be emphasised in some other way. If it is a rule that subtitles are italicised, the scientific name should not be italicised.

L480 - Neisseria gonorrhoeae.

L681 – L682 - Please see previous comments.

L693 – S. aureus (in italic).

Author Response

Dear Reviewer 4,

Thank you very much for the time allocated for reviewing the MS- "Current State of Knowledge Regarding WHO High Priority Pathogens – Resistance Mechanisms and Proposed Solutions through Candidates such as Essential Oils: A systematic review "  as well as for your pertinent observations.

Please see below our response to your comments:

“AMR is a global health problem that WHO will address in 2021. Effective prophylactic measures and more effective and non-toxic antimicrobials are needed. Essential oils (EOs) are still newcomers when it comes to treating infections in the clinical setting, as there is insufficient data on their in vivo activity and toxicity. In this review, the authors address the concept of antibiotic resistance and its major determinants, the methodology used to address the problem, and the current state of research on the pathogenesis, mechanism of resistance and activity of various essential oils against six priority pathogens. The article is comprehensive, well written and very enlightening. However, in some parts the article seems to be tailored to the problem in Europe, especially Eastern Europe. Given the importance of the topic, it would be ideal if the authors made an effort to problematise the issue globally. Please, verify the specifics comments bellow.

L44 – Campylobacter (in italic);

L263-264 - Is there a problem with mentioning the name of the microbiologist in this case?

Fig. 1 - Congratulations on this figure. Very instructive for applied methodology.

L269 - I think that in these cases the scientific name should be emphasised in some other way. If it is a rule that subtitles are italicised, the scientific name should not be italicised. L337 (idem).

L338 - For stylistic reasons, it is advisable not to begin a sentence with the abbreviation of a name, even in the case of generic names. In this case, rather write the generic name without the abbreviation.

L479 - I think that in these cases the scientific name should be emphasised in some other way. If it is a rule that subtitles are italicised, the scientific name should not be italicised.

L480 - Neisseria gonorrhoeae.

L681 – L682 - Please see previous comments.

L693 – S. aureus (in italic). »

Response

We thank the reviewer for the suggestions and have made modifications to address the raised issues.

Our introductory section now contains a more comprehensive part regarding the global situation of AMR. We are confident that the worldwide magnitude of antimicrobial resistance is clearer in this present form of the manuscript.

We made adjustments to the style of the text and checked that the scientific names of bacterial strains were correctly placed.

We have also mentioned the initials of the microbiologist.

Sincerely,

Cl. pharm. Valentina Buda
Associate Professor
Discipline of Clinical pharmacy, Communication in pharmacy and Pharmaceutical care

Faculty of Pharmacy
"Victor Babes" University of Medicine and Pharmacy, Timisoara, Romania

Reviewer 5 Report

This is an interesting manuscript, with several strengths. This article fits the scope of the journal, the abbreviations are usual and explained, the method section is well written, the future research possibilities are provided and provides more data about antimicrobial resistance. In order to straighten this article, points should be addressed accordingly before consideration in the journal:

Suggestions:

- The introduction section is too long, it should be shortened to 2 pages or 3 with the table. 

Author Response

Dear Reviewer 5,

Thank you very much for the time allocated for reviewing the MS- "Current State of Knowledge Regarding WHO High Priority Pathogens – Resistance Mechanisms and Proposed Solutions through Candidates such as Essential Oils: A systematic review "  as well as for your pertinent observations.

Please see below our response to your comment:

“This is an interesting manuscript, with several strengths. This article fits the scope of the journal, the abbreviations are usual and explained, the method section is well written, the future research possibilities are provided and provides more data about antimicrobial resistance. In order to straighten this article, points should be addressed accordingly before consideration in the journal:

Suggestions:

- The introduction section is too long, it should be shortened to 2 pages or 3 with the table.“

Response

We appreciate the reviewer’s comments and have made proper adjustments to our manuscript. The introduction has been compressed to be more succinct. Here, we also modified the 1.3. section to switch the focus to the global antimicrobial resistance situation. Moreover, to allow the reader to better understand the technical information within tables, we provided extra details regarding the methods used to assess the antimicrobial activity of essential oils (as other reviewer suggested).

Sincerely,

Cl. pharm. Valentina Buda
Associate Professor
Discipline of Clinical pharmacy, Communication in pharmacy and Pharmaceutical care

Faculty of Pharmacy
"Victor Babes" University of Medicine and Pharmacy, Timisoara, Romania

Round 2

Reviewer 3 Report

The authors provided a revised manuscript addressing all reviewers' comments and suggestions in a satisfactory manner.